# Associations of Prenatal and Postnatal Maternal Depressive Symptoms with Offspring Cognition and Behavior in Mid-Childhood: A Prospective Cohort Study

**DOI:** 10.3390/ijerph16061007

**Published:** 2019-03-20

**Authors:** Sabrina Faleschini, Sheryl L. Rifas-Shiman, Henning Tiemeier, Emily Oken, Marie-France Hivert

**Affiliations:** 1School of Psychology, Laval University, Quebec, QC G1V 0A6, Canada; sabrina.faleschini.1@ulaval.ca; 2Division of Chronic Disease Research Across the Lifecourse, Department of Population Medicine, Harvard Medical School and Harvard Pilgrim Health Care Institute, Boston, MA 02215, USA; sheryl_rifas@harvardpilgrim.org (S.L.R.-S.); emily_oken@harvardpilgrim.org (E.O.); 3Department of Social and Behavioral Sciences, Harvard T.H. Chan School of Public Health, Boston, MA 02115, USA; tiemeier@hsph.harvard.edu; 4Department of Nutrition, Harvard T.H. Chan School of Public Health, Boston, MA 02115, USA; 5Diabetes Unit, Massachusetts General Hospital, Boston, MA 02114, USA

**Keywords:** maternal depression, child development, behavior problems

## Abstract

Exposure to maternal depressive symptoms in the peri-pregnancy periods may be associated with poorer child development, but research is often limited to only maternal assessments of behavior and cognition. This study investigates the specific periods of prenatal and postnatal exposure to maternal depressive symptoms in association with child development using reports from teachers and mothers. This study is based on 1225 mother–child pairs from Project Viva, a prospective pre-birth cohort study. Mothers reported depressive symptoms on the Edinburgh Postpartum Depression Scale (EPDS) in mid-pregnancy as well as at 6 months and 12 months postpartum. Teachers and mothers reported child executive functions using the Behavioral Rating Inventory of Executive Function (BRIEF) and behavior using the Strengths and Difficulties Questionnaire (SDQ). Children completed the Kaufman Brief Intelligence Test (KBIT-2), the Wide Range Assessment of Visual Motor Abilities (WRAVMA), and the Visual Memory Index of the Wide Range Assessment of Memory and Learning (WRAML). We used multivariable linear regression models to examine associations of prenatal and postpartum depressive symptoms with outcomes. Many of the crude associations observed were attenuated after adjusting for demographic factors and maternal IQ, yet some remained significant. For example, high prenatal maternal depressive symptoms were associated with poorer scores on the BRIEF Behavior Regulation Index and some SDQ scales based on reports from teachers and mothers. High prenatal maternal depressive symptoms were associated with poorer behavioral development. Postpartum symptoms did not show strong associations with outcomes once we adjusted for the prenatal period.

## 1. Introduction

Pregnancy and the postpartum are periods during which women are at higher risk of experiencing depressive symptoms [1,2]. Early-life exposure of children to maternal depressive symptoms, during in utero development or in the first years of life, could have long-lasting consequences [3,4]. Prenatal exposure to an adverse environment may have an effect on fetal and child development through physiological intrauterine mechanisms, while the postnatal environment may contribute through the influence of parental behaviors and care [5,6,7,8].

Some studies have suggested that children exposed to prenatal or postnatal maternal depressive symptoms are more likely to present deficits in cognitive and behavioral development than unexposed children [9,10,11,12,13,14,15]. However, these studies are not without limitations. First, studies investigating pre- and/or post-natal maternal depressive symptoms associations with child development often failed to control for maternal IQ, which is known to be related to child cognitive development and executive function [16]. Moreover, studies often use the mother’s behavioral evaluation of their own child as a reference for child development, which can lead to an overestimation of behavior problems, especially from women with higher depressive symptoms. Thus, this non-random bias requires non-maternal sources of information to estimate child development [17,18]. 

To address these gaps, this study aimed to investigate the specific periods of prenatal and postnatal exposure to maternal depressive symptoms associations with behavioral and cognitive development among school-age children. Using a large prospective longitudinal study from pregnancy to mid-childhood, teachers’ reports were used as the primary outcome for the behavioral evaluation in addition to mothers’ reports.

## 2. Materials and Methods

### 2.1. Population and Study Design

Project Viva is a prospective pre-birth cohort of mother and child pairs enrolled between 1999 and 2002 at initial prenatal visits at Atrius Harvard Vanguard Medical Associates, a multispecialty group practice in eastern Massachusetts. Exclusion criteria included multiple gestation, inability to answer questions in English, gestational age ≥22 weeks at recruitment, and plans to move away from the study area before delivery. All participating women provided written informed consent, and institutional review boards reviewed and approved the project in line with ethical standards established by the Declaration of Helsinki. A full description of the study, including recruitment and retention details, has been previously published [19]. 

Of 2128 live singleton infants and their mothers, we excluded 903 who did not attend a mid-childhood visit, resulting in 1225 pairs included in this analysis. Compared with the 1225 included mothers, the 903 excluded were slightly younger at enrollment (31.3 vs. 32.2 years) and were less likely to report white race/ethnicity (64% vs. 68%), be college graduates (59% vs. 69%), and to have a household income greater than $70,000 USD/year (55% vs. 60%).

### 2.2. Measures

#### 2.2.1. Maternal Depressive Symptoms

Mothers reported their depressive symptoms using the Edinburgh Postpartum Depression Scale (EPDS) in mid-pregnancy as well as at 6 months and 12 months postpartum [20]. The questionnaire is a 10-item scale and answers are on a 4-point Likert scale from 0 to 3 for a possible range of 0 to 30; a higher score indicates a higher level of depressive symptoms. The EPDS does not allow a clinical diagnosis of depression, but a score ≥13 indicates probable depression [21]. The scale has a sensitivity of 86% and a specificity of 78% for the diagnosis of depression [20]. The EPDS was initially made for postpartum depression but has been frequently used to evaluate prenatal depressive symptoms as well [22,23]. Internal consistency in our sample was high, with Cronbach’s alpha of 0.86 at mid-pregnancy, 0.87 at 6 months, and 0.86 at 12 months postpartum, similar to other studies [24,25].

#### 2.2.2. Child Behavioral Outcomes

At mid-childhood, teachers and mothers completed two validated behavioral rating scales of child participants: The Behavior Rating Inventory of Executive Function (BRIEF) and the Strengths and Difficulties Questionnaire (SDQ). The BRIEF evaluates executive function, assessing behaviors including planning and organization, working memory, inhibition of inappropriate impulses, emotional control, and ability to re-evaluate and shift problem solving approaches, and is validated and standardized for use in children aged 5 to 18 [26]. Trained research staff scored completed BRIEF questionnaires according to published guidelines to generate two sub-scores (Metacognition Index (MI) and Behavioral Regulation Index (BRI)), and one overall Global Executive Composite score (GEC), which incorporates both the MI and the BRI. The MI, BRI, and GEC scores were each standardized to mean = 50, standard deviation (SD) = 10 using published reference data, and higher scores represent greater problems [26]. The SDQ assesses behaviors in five categories (prosocial, hyperactivity, emotional problems, conduct problems, and peer problems) [27], and has good agreement with the Child Behavior Checklist [28]. It is frequently used in research and clinical settings and is valid and reliable among children aged 4 to 16 [29]. Trained Project Viva staff scored the SDQ questionnaires, yielding sub-scores in each behavioral category and a measure of total behavioral difficulties (possible scores range from 0 to 40 with higher scores representing greater problems).

#### 2.2.3. Child Cognitive Outcomes

In mid-childhood, staff assessed verbal and non-verbal intelligence using the Kaufman Brief Intelligence Test, Second Edition (KBIT-2) [30]; visual-motor skills using the Wide Range Assessment of Visual Motor Abilities (WRAVMA) drawing subtest; and visual memory (design memory and picture memory) using the Visual Memory Index of the Wide Range Assessment of Memory and Learning, Second Edition (WRAML) [31]. The KBIT-2 is a valid and reliable measure for children and adults aged 4 to 90, standardized using a representative U.S. sample. KBIT-2 scores are moderately to highly correlated with relevant subscores on the Wechsler Intelligence Scale for Children (WISC-III) (*r* = 0.76 for the KBIT-2 IQ composite and WISC-III full-scale IQ) [30]. The WRAML is standardized for ages 5 to 90 based on a representative U.S sample; among children aged 6 to 16, the WRAML Visual Memory Index showed moderate correlation with relevant indices from the Children’s Memory Scale (*r* = 0.48 for General Memory, 0.52 for Attention/Concentration) [31]. Scaled scores were standardized for age to mean = 100, SD = 15 for the KBIT-2 and WRAVMA, and mean = 10, SD = 3 for WRAML design memory and picture memory subscores, using published reference data [30,31].

#### 2.2.4. Other Measures

Using a combination of self-administered questionnaires and interviews during pregnancy, we collected information about maternal race/ethnicity, age, education, pre-pregnancy weight, height, and smoking. We assessed maternal intelligence at the mid-childhood visit using the KBIT-2 [30]. We obtained child sex, birth weight, and date of birth from hospital records, and determined gestational age at birth from last menstrual period, or prenatal ultrasound if the two estimates differed by more than 10 days. We calculated birth weight for gestational age z-score as a measure of fetal growth [32]. At 6 and 12 months postpartum, we assessed duration of any breastfeeding which has been shown to be associated with cognitive outcomes in this cohort [33]. We measured maternal pregnancy-related anxiety using the Pregnancy-Related Anxiety Scale [34] and we assessed prescription of antidepressant during pregnancy.

#### 2.2.5. Statistical Analyses

We performed a series of multivariable regression models to evaluate associations of period-specific (mid-pregnancy, 6 months postpartum, 12 months postpartum) exposure to high maternal depressive symptoms (EPDS score ≥13), with cognitive and behavioral outcomes in mid-childhood. For the mid-pregnancy maternal exposure, Model 1 was unadjusted, and Model 2 was adjusted for potential confounders, including maternal race/ethnicity, age at enrollment, education, household income, pre-pregnancy body mass index (BMI), smoking during pregnancy, and child sex. Model 3 was additionally adjusted for maternal IQ (KBIT-2). For the 6-month and 12-month postpartum exposures, Model 1 was unadjusted, Model 2 was adjusted for potential confounders, including maternal race/ethnicity, age at enrollment, education, household income, pre-pregnancy BMI, smoking during pregnancy, child sex, gestation length, birth weight/gestational age z-score, and breastfeeding duration, Model 3 was additionally adjusted for maternal IQ (KBIT-2), and Model 4 was additionally adjusted for depressive symptoms at prior periods assessed in this study (mid-pregnancy EPDS ≥13 for 6-month models, and pre-pregnancy plus 6-month EPDS ≥13 for 12-month models). To account for missing data, we performed multiple imputation for all 2128 mother–child pairs in Project Viva. We then limited the analysis to the 1225 included participants. We used SAS (Proc MI) to impute 50 values for each missing observation and combined multivariable modeling estimates using Proc MI ANALYZE in SAS. An alternative approach, using only participants with complete data, yielded similar results thus we are presenting the MI for optimal sample sizes. We performed all analyses in SAS version 9.4 (SAS Institute Inc., Cary, NC, USA). 

## 3. Results

### 3.1. Participants

The mean (SD) age of women at study enrollment was 32.2 years (5.2); 68.3% were white, and 68.9% were college graduates. Overall, 10.0% had high depressive symptoms (EPDS ≥13) in mid-pregnancy, 9.4% at 6 months postpartum, and 7.4% at 12 months postpartum (Table 1). Compared with women with EPDS <13 in mid-pregnancy, women who had EPDS ≥13 in mid-pregnancy were less likely to report white race/ethnicity (49.1% vs. 70.4%), be college graduates (57.2% vs. 70.2%), or have a household income greater than $70,000 USD/year (38.0% vs. 62.5%).

### 3.2. Exposure to Depressive Symptoms during the Prenatal Period

Exposure to high maternal depressive symptoms in mid-pregnancy (EPDS ≥13) was associated with poorer scores on behavioral executive function based on the three BRIEF subscales, according to both teachers’ and mothers’ reports (Table 2 and Appendix A). After accounting for confounders and maternal IQ, the association was attenuated, yet remained significant on the BRIEF Behavior Regulation Index with 2.44 points higher based on teachers’ report (95% CI (0.00, 4.88); Table 2). Prenatal depressive symptoms were associated with poorer behavioral SDQ outcomes according to teachers’ reports, reflected as about 1.79 points (95% CI (0.43, 3.14)) greater on the SDQ Total Difficulties scale (Table 2). When adjusting for potential confounders and maternal IQ, this association was attenuated. Associations between high maternal depressive symptoms in mid-pregnancy and behavioral outcomes in mid-childhood reported by mothers showed very similar results according to the BRIEF scores and slightly larger effect sizes for assessment by SDQ (Appendix A).

Regarding cognitive development, having high maternal depressive symptoms in mid-pregnancy was associated with a lower KBIT-2 Verbal subscale (−4.69 points, 95% CI (−8.01, −1.36)) in the unadjusted model (Table 2, Model 1) but this was attenuated when adjusting for confounders and for maternal IQ. We did not find associations with any other subscales of cognitive assessment.

Furthermore, we conducted sensitivity analyses adjusting for prenatal fetal exposure to antidepressants or maternal pregnancy-related anxiety as additional covariates and neither changed the results substantially, so we kept our primary analyses as described above.

### 3.3. Exposure to Depressive Symptoms during the Postnatal Period

High maternal depressive symptoms (EPDS ≥13) at 6 months postpartum was associated with higher scores on the BRIEF Global Executive Composite score (2.78 points, 95% CI (0.18, 5.39)) and on the BRIEF Metacognition Index (2.86 points, 95% CI (0.21, 5.50)), according to teachers’ reports (Appendix A). After adjusting for confounders, the associations were greatly attenuated, especially after accounting for prenatal depressive symptoms. Teachers’ report of SDQ Total Difficulties was also 1.46 points higher (95% CI (0.07, 2.86)) in children exposed to high maternal depressive symptoms at 6 months postpartum, but was much reduced after accounting for confounders (Appendix A). Associations with scores provided by mothers’ report provided similar patterns. High maternal depressive symptoms exposure at 6 months postpartum was not associated with child cognitive development.

Exposure to maternal depressive symptoms at 12 months postpartum was associated with poorer behavioral executive function on the three BRIEF subscales according to teachers’ report in the unadjusted models with a mean of 4.22 points higher (Appendix A). Teachers’ report of SDQ Total Difficulties showed a greater score of 2.14 points (95% CI (0.47, 3.81)). However, when adjusting for confounders the associations were substantially reduced with wide confidence intervals. Results by mothers’ reports provided similar patterns. High maternal depressive symptoms at 12 months postpartum were not associated with overall cognitive development after adjusting for confounders except for lower WRAML Picture Memory (–0.93 points, 95% CI (−1.79, −0.08)).

## 4. Discussion

This study examined the timing of exposure to high maternal depressive symptoms in regard to child behavioral and cognitive development using teachers’ report, offering a more objective behavioral evaluation compared to mothers’ report alone. We found that exposure to high prenatal maternal depressive symptoms predicted poorer behavioral executive functions and higher levels of behavioral difficulties in children. In contrast, we found little evidence of associations between maternal depressive symptoms at 6 and 12 months postpartum and mid-childhood development, after accounting for confounders and depressive symptoms in the prenatal period. For any of exposure periods tested, our crude estimates of associations were attenuated by adjusting our models for socio-demographic characteristics and maternal IQ, highlight the importance of taking into account these confounders.

Associations of maternal depressive symptoms at different time points with cognitive and behavioral development have been studied mainly among infants and preschoolers, and previous studies generally use only mother report of child development or do not control for maternal IQ, which can influence the results [35,36,37,38,39]. In Project Viva, we have previously observed a lack of associations between prenatal maternal depression and cognitive development at 3 years of age but the long-term effect was unknown [40]. In the current study, higher prenatal maternal depressive symptoms predicted poorer behavioral executive function in school-aged children using teachers’ and mothers’ reports, extending our prior report and addressing some of the limitations of other previous studies. In contrast, we did not find associations with exposure to high maternal depressive symptoms during the postnatal period, after accounting for potential confounders and depressive symptoms in previous periods. Our results are in line with previous research, which has found a long-term effect of prenatal exposure to mood disorders on the development of infants [12,35,41,42,43,44]. This is consistent with the hypothesis of fetal programming in which in utero development may be compromised when the fetus is exposed to maternal mood disorders during a highly sensitive period of time [45,46]. Our study adds to the current literature by providing child behavioral assessment by an adult other than the mother herself, which has been shown to incorporate differential bias for mothers with history of high depressive symptoms.

The present study should be interpreted in the context of some limitations. The EPDS is an instrument used to assess depressive symptoms but is not equivalent to clinical diagnosis, yet it has been shown to compare well with longer-term depressive symptoms measures and with clinically diagnosed depression [47]. Also, depressive symptoms can be associated with other psychopathologies; thus, we cannot exclude the potential effect of related mental illness or high levels of daily stress on child development. Moreover, unlike maternal IQ, we did not measure maternal executive function or behavior problems, which may be part of the confounding structure. We cannot exclude that effects found on prenatal exposure could include influences from postnatal symptoms, but we choose not to control for postnatal symptoms in the statistical model as they could be part of the pathways. Finally, we are fully aware that mothers’ reports of child development may be biased, especially in the case of depressed mothers [17], so all results from mothers’ report models should be interpreted with caution. This is why we chose to focus on teachers’ reports. This study also has many strengths. To untangle the effects of prenatal and postnatal exposure to maternal depressive symptoms and to examine persistent effect across time, this study used longitudinal data regarding the effects of the timing of exposure to maternal depressive symptoms among school-age children simultaneously on behavioral and cognitive development. To provide a more objective view of child executive function and behavioral development, we chose teachers’ evaluation as our primary outcomes. We also adjusted our analyses for a broad range of confounders known to influence depressive symptoms or child developmental outcomes including the effect of maternal IQ.

## 5. Conclusions

The current study highlights the associations of prenatal maternal depressive symptoms in children’s behavioral development in multiple domains, yet the impact seems limited to behavioral regulation of executive functions based on teachers’ assessment of behaviors in mid-childhood. It is reassuring that we found little evidence (and if so of small magnitude) of effects of exposure to high depressive symptoms (either pre or post-natal) on mid-childhood cognition after adjusting for demographic factors and maternal IQ. To contribute to a better understanding of the mechanisms explaining the association between prenatal depressive symptoms and behavior in mid-childhood, further studies should include a physiological evaluation of response to high depressive symptoms in women during pregnancy. Depression in women is a major public health problem and in recent years public health campaigns have rightfully directed attention towards depression in the postpartum period. The current study however highlights the crucial role of prenatal depression on later child development. Therefore, public health campaigns should also extend their focus to raise public awareness about mental health during the pregnancy period to improve the health of both mothers and their future children.

## Figures and Tables

**Table 1 ijerph-16-01007-t001:** Participants’ characteristics according to depressive symptoms at mid-pregnancy.

Participants’ Characteristics and Outcomes	Overall	EPDS <13	EPDS ≥13
	(*n* = 1225)	(*n* = 1103)	(*n* = 122)
	Mean (SD) or %		
**Maternal characteristics**			
Age at enrollment, years	32.2 (5.2)	32.4 (5.1)	30.3 (5.9)
Race/ethnicity, %			
Black	15.5	14.2	27.9
Hispanic	6.3	5.7	11.8
White	68.3	70.4	49.1
Other	9.9	9.7	11.1
College degree, %	68.9	70.2	57.2
IQ (KBIT-2 composite), points	106.5 (15.4)	106.9 (15.2)	103.4 (16.6)
Household income > $70,000 USD/year, %	60.1	62.5	38.0
Antidepressant prescribed during pregnancy, %	2.8	2.0	10.2
Pre-pregnancy BMI, kg/m^2^	24.7 (5.1)	24.6 (5.1)	25.2 (5.4)
Pregnancy smoking status, %			
Never	71.3	71.2	72.7
Former	19.0	19.9	11.0
During pregnancy	9.7	8.9	16.3
EPDS ≥13 at 6 months postpartum, %	9.4	6.6	34.9
EPDS ≥13 at 12 months postpartum, %	7.4	5.2	27.1
**Child characteristics**			
Age mid-childhood visit, years	7.9 (0.8)	7.9 (0.8)	8.0 (0.9)
Sex, male %	50.3	50.8	45.8
Gestational age, weeks	39.5 (1.8)	39.5 (1.8)	39.3 (1.8)
Birth weight/gestational age z-score	0.19 (0.97)	0.21 (0.96)	0.03 (1.02)
Duration of any breastfeeding at 6 months, in months	4.1 (2.4)	4.1 (2.4)	3.9 (2.5)
Duration of any breastfeeding at 12 months, in months	6.3 (4.6)	6.3 (4.6)	6.1 (4.8)
**Behavior and executive function in mid-childhood**			
Teacher-rated			
BRIEF Global Executive Composite	51.2 (10.5)	50.8 (10.0)	54.6 (13.2)
BRIEF Behavior Regulation Index	50.8 (10.2)	50.4 (9.7)	54.3 (13.3)
BRIEF Metacognition Index	51.3 (10.8)	50.9 (10.5)	54.2 (12.9)
SDQ Total Difficulties	6.4 (5.8)	6.2 (5.6)	8.0 (7.3)
SDQ Prosocial	8.0 (2.2)	8.1 (2.2)	7.9 (2.4)
SDQ Hyperactivity	3.1 (3.0)	3.0 (3.0)	3.6 (3.3)
SDQ Emotional Problems	1.3 (1.8)	1.3 (1.7)	1.8 (2.1)
SDQ Conduct Problems	0.9 (1.6)	0.8 (1.5)	1.3 (2.0)
SDQ Peer Problems	1.1 (1.6)	1.1 (1.6)	1.3 (1.9)
Mother-rated			
BRIEF Global Executive Composite	48.7 (9.2)	48.4 (9.0)	51.0 (10.4)
BRIEF Behavior Regulation Index	48.2 (8.9)	48.0 (8.6)	50.7 (10.3)
BRIEF Metacognition Index	48.5 (8.8)	48.3 (8.7)	50.3 (9.4)
SDQ Total Difficulties	6.6 (4.8)	6.3 (4.6)	9.0 (6.0)
SDQ Prosocial	8.5 (1.7)	8.5 (1.7)	8.3 (1.8)
SDQ Hyperactivity	2.9 (2.4)	2.9 (2.4)	3.5 (2.5)
SDQ Emotional Problems	1.6 (1.7)	1.5 (1.7)	2.3 (2.0)
SDQ Conduct Problems	1.0 (1.3)	1.0 (1.2)	1.4 (1.6)
SDQ Peer Problems	1.0 (1.4)	1.0 (1.3)	1.7 (1.8)
**Offspring cognition in mid-childhood**			
KBIT-2 Verbal	111.9 (15.0)	112.4 (14.8)	107.7 (16.4)
KBIT-2 Non-Verbal	106.3 (16.9)	106.6 (16.8)	103.8 (17.4)
WRAVMA Visual–Motor	91.9 (16.8)	92.0 (16.7)	91.4 (16.9)
WRAML Design Memory	8.0 (2.8)	8.0 (2.8)	7.7 (3.1)
WRAML Picture Memory	8.9 (3.0)	8.9 (3.0)	9.3 (3.3)
WRAML Visual Memory, Global Score	16.9 (4.4)	16.9 (4.3)	16.9 (4.7)

Note: EPDS = Edinburgh Postnatal Depression Scale (range 0–30); BRIEF = Behavior Rating Inventory of Executive Function (standardized mean = 50, SD = 10); SDQ = Strengths and Difficulties Questionnaire (Total Difficulties range 0–40 and subscales range 0–10); KBIT-2 = Kaufman Brief Intelligence Test (standardized mean = 100, SD = 15); WRAML = Wide Range Assessment of Memory and Learning (standardized mean = 10, SD = 3); WRAVMA = Wide Range Assessment of Visual Motor Abilities (standardized mean = 100, SD = 15); SD = standard deviation. Higher scores on the BRIEF and SDQ indicate greater behavior problems.

**Table 2 ijerph-16-01007-t002:** Associations of maternal depressive symptoms in mid-pregnancy with offspring behavioral and cognitive outcomes in mid-childhood in the Project Viva cohort (*n* = 1225).

Children Developmental Outcomes	Model 1	Model 2	Model 3
	*β* (95% CI)		
**Behavior and executive function—Teacher-rated**			
BRIEF Global Executive Composite	3.83 (1.32, 6.35)	2.11 (−0.30, 4.52)	2.20 (−0.21, 4.60)
BRIEF Behavior Regulation Index	3.93 (1.41, 6.45)	2.46 (0.02, 4.89)	2.44 (0.00, 4.88)
BRIEF Metacognition Index	3.26 (0.72, 5.81)	1.59 (−0.89, 4.06)	1.72 (−0.74, 4.18)
SDQ Total Difficulties	1.79 (0.43, 3.14)	1.27 (−0.05, 2.59)	1.30 (−0.02, 2.62)
SDQ Prosocial	−0.13 (−0.64, 0.37)	−0.07 (−0.57, 0.43)	−0.07 (−0.57, 0.43)
SDQ Hyperactivity	0.58 (−0.10, 1.27)	0.38 (−0.28, 1.04)	0.41 (−0.25, 1.07)
SDQ Emotional Problems	0.48 (0.04, 0.92)	0.42 (−0.02, 0.87)	0.43 (−0.02, 0.87)
SDQ Conduct Problems	0.44 (0.06, 0.83)	0.24 (−0.14, 0.62)	0.25 (−0.14, 0.63)
SDQ Peer Problems	0.28 (−0.08, 0.64)	0.23 (−0.14, 0.59)	0.22 (−0.15, 0.59)
**Cognition**			
KBIT-2 Verbal	−4.69 (−8.01, −1.36)	−0.68 (−3.61, 2.25)	−1.28 (−4.10, 1.54)
KBIT-2 Non-Verbal	−2.82 (−6.72, 1.09)	−1.15 (−5.01, 2.71)	−1.60 (−5.41, 2.22)
WRAVMA Visual–Motor	−0.61 (−4.31, 3.09)	0.15 (−3.57, 3.88)	−0.02 (−3.75, 3.70)
WRAML Design Memory	−0.35 (−0.98, 0.27)	−0.15 (−0.79, 0.48)	−0.19 (−0.82, 0.43)
WRAML Picture Memory	0.39 (−0.26, 1.05)	0.54 (−0.13, 1.20)	0.53 (−0.14, 1.20)
WRAML Visual Memory, Global Score	0.04 (−0.90, 0.97)	0.38 (−0.57, 1.33)	0.34 (−0.61, 1.29)

Note: Model 1 = Unadjusted; Model 2 = Adjusted for maternal race/ethnicity, age at enrollment, education, household income, pre-pregnancy BMI, smoking during pregnancy, and child sex; Model 3 = Model 2 additionally adjusted for maternal IQ (KBIT-2).

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
