# Peer review of "Associations of Prenatal and Postnatal Maternal Depressive Symptoms with Offspring Cognition and Behavior in Mid-Childhood: A Prospective Cohort Study"

_ijerph, 2019, doi:10.3390/ijerph16061007_

Round 1
Reviewer 1 Report
The manuscript is well written and the topic of significant importance. However, there are two things that I find missing and would like to see in the manuscript.
According to table 1 a significantly lower percentage of mothers with depression have income above 70,000 compared to the ones without depression. Yet, income is not included as a confounding factor in the analyses. Can you please explain why and i would encourage you to include it in the analyses, because income has been found associated with depressive symptoms in mothers and behavioural changes in children.
The IJERPH is a journal that specializes in public health, yet i don't see any public health implications included in the discussion. What is your view on the problem as a public health issue? Can you give examples if strategies that can be applied to address the problem such as successful public health prevention strategies, or treatment strategies? I would like to see a bit more of your view on the topic in the discussion.
Author Response
We thank you and the reviewers for the useful comments and suggestions. We have carefully addressed the concerns of the reviewers. Specifically, we conducted analysis including household income as an additional co-variable in our models. The inclusion of this covariate did not significantly change our results, but we are glad to present our revised manuscript with updated results that included this additional co-variable in our multi-adjusted models. We now present the updated results in blue font in the manuscript and in supplementary material. Moreover, we addressed the concern about implication of maternal depression in a public health point of view in the manuscript.
We are grateful for this opportunity to clarify and to improve our manuscript. We have included a point-by-point response to the reviewers and we have adjusted the manuscript using the track-change function.
Comments
Reviewer: 1
Comments to the author:
The manuscript is well written and the topic of significant importance. However, there are two things that I find missing and would like to see in the manuscript.
Response: We thank the reviewer's for his/her positive comments and useful suggestions below.
1. According to table 1 a significantly lower percentage of mothers with depression have income above 70,000 compared to the ones without depression. Yet, income is not included as a confounding factor in the analyses. Can you please explain why and i would encourage you to include it in the analyses, because income has been found associated with depressive symptoms in mothers and behavioral changes in children.
Response: We thank the reviewer for the opportunity to improve our manuscript. We performed new analysis including household income as a covariate in our models. Including household income did not change significantly our results, thus we now present the updated results in blue font in the manuscript (text and Table 2) and in supplementary material.
2. The IJERPH is a journal that specializes in public health, yet i don't see any public health implications included in the discussion. What is your view on the problem as a public health issue? Can you give examples if strategies that can be applied to address the problem such as successful public health prevention strategies, or treatment strategies? I would like to see a bit more of your view on the topic in the discussion.
Response: We thank the reviewer for the useful suggestion. We have included public health implications on page 8 lines 266-271.
“Depression in women is a major public health problem and in recent years public health campaigns have rightfully directed attention towards depression in the postpartum period. The current study however highlights the crucial role of prenatal depression on later child development. Therefore public health campaigns should also extend their focus to raise public awareness about mental health during the pregnancy period to improve health of both mothers and their future children.”

Reviewer 2 Report
A clear, well constructed manuscript. Care should be taken to ensure conclusions highlighted in the abstract and at the end of the manuscript acknowledge those effects attenuated in adjusted models when controlling for demographics and maternal IQ (Model 2 and 3). Otherwise no major concerns are noted.
Author Response
We thank you and the reviewers for the useful comments and suggestions. We have carefully addressed the concerns of the reviewers. Specifically, we conducted analysis including household income as an additional co-variable in our models. The inclusion of this covariate did not significantly change our results, but we are glad to present our revised manuscript with updated results that included this additional co-variable in our multi-adjusted models. We now present the updated results in blue font in the manuscript and in supplementary material. Moreover, we addressed the concern about implication of maternal depression in a public health point of view in the manuscript.
We are grateful for this opportunity to clarify and to improve our manuscript. We have included a point-by-point response to the reviewers and we have adjusted the manuscript using the track-change function.
Comments
Reviewer: 2
A clear, well constructed manuscript. Care should be taken to ensure conclusions highlighted in the abstract and at the end of the manuscript acknowledge those effects attenuated in adjusted models when controlling for demographics and maternal IQ (Model 2 and 3). Otherwise no major concerns are noted.
Response: We thank the reviewer for the useful suggestion. We have modified the abstract (page 1) and the discussion and conclusion (page 7 and 8) to highlight this aspect of our findings.
